# Clinical Experience with Autofluorescence Guided Oral Squamous Cell Carcinoma Surgery

**DOI:** 10.3390/diagnostics13203161

**Published:** 2023-10-10

**Authors:** Petr Pošta, Andreas Kolk, Kristýna Pivovarčíková, Jan Liška, Jiří Genčur, Omid Moztarzadeh, Christos Micopulos, Adam Pěnkava, Maria Frolo, Oliver Bissinger, Lukáš Hauer

**Affiliations:** 1Department of Stomatology, University Hospital Pilsen, Faculty of Medicine, Charles University, 32300 Pilsen, Czech Republic; liskaj@fnplzen.cz (J.L.); hauerl@fnplzen.cz (L.H.); 2Department of Oral and Maxillofacial Surgery, Medical University of Innsbruck, 6020 Innsbruck, Austria; andreas.kolk@i-med.ac.at (A.K.); oliver.bissinger@t-online.de (O.B.); 3Sikl’s Department of Pathology, Faculty of Medicine, Charles University, 32300 Pilsen, Czech Republic; pivovarcikovak@fnplzen.cz; 4Bioptic Laboratory Ltd., 32600 Pilsen, Czech Republic; 5Department of Anatomy, Faculty of Medicine, Charles University, 32300 Pilsen, Czech Republic

**Keywords:** autofluorescence, oral squamous cell carcinoma, surgical treatment, margin status

## Abstract

In our study, the effect of the use of autofluorescence (Visually Enhanced Lesion Scope—VELscope) on increasing the success rate of surgical treatment in oral squamous carcinoma (OSCC) was investigated. Our hypothesis was tested on a group of 122 patients suffering from OSCC, randomized into a study and a control group enrolled in our study after meeting the inclusion criteria. The preoperative checkup via VELscope, accompanied by the marking of the range of a loss of fluorescence in the study group, was performed before the surgery. We developed a unique mucosal tattoo marking technique for this purpose. The histopathological results after surgical treatment, i.e., the margin status, were then compared. In the study group, we achieved pathological free margin (pFM) in 55 patients, pathological close margin (pCM) in 6 cases, and we encountered no cases of pathological positive margin (pPM) in the mucosal layer. In comparison, the control group results revealed pPM in 7 cases, pCM in 14 cases, and pFM in 40 of all cases in the mucosal layer. This study demonstrated that preoperative autofluorescence assessment of the mucosal surroundings of OSCC increased the ability to achieve pFM resection 4.8 times in terms of lateral margins.

## 1. Introduction

Oral squamous cell carcinoma is a serious and relatively frequent disease of the oral cavity and, unfortunately, also belongs among the most common malignancies in the orofacial region [1]. The estimated age-standardized rate of incidence in Europe is 16.9, and in the European Union it is 17.0, followed by a mortality of 7.1 and 6.7, respectively, per 100,000 for the year 2020 [2]. The situation in the Czech Republic is comparable to the EU average ([Fig diagnostics-13-03161-ch001]). Surgical treatment, especially in the early stages of the disease, has the best curative results [3]. The radicality of the procedure is crucial for prognosis and treatment success [4,5,6,7,8,9,10,11]. It is essential to determine the area of tissue affected by tumor cells from healthy cells during surgery to ensure the radicality.

The presence of clinical occult malignant transformation of mucosal cells, which is not noticeable during surgery, is a frequent source of local recurrence [12]. The lateral occult extension of the tumor varies in size and is irregular, so the commonly used strategy of a 10 mm safe margin of healthy tissue around the visible tumor is not effective in achieving a tumor-free surgical margin [13]. New effective examination and imaging techniques are being developed that allow the surgeon to better visualize the boundaries of the primary tumor before and during the surgery more precisely [14,15,16,17,18,19,20,21,22,23,24,25,26]. With the help of these techniques, it is possible, among other things, to have a positive impact on margin surveillance. An ideal investigative technique for this purpose should meet certain characteristics. High sensitivity and specificity, while maintaining complete non-invasiveness, are essential. Furthermore, it is important that this examination is feasible intraoperatively, achieves stable results, is reproducible, quick and simple, as well as economically sustainable and applicable in a wide range of practice. Optical methods fulfill most of these properties, but so far, they are burdened with a number of shortcomings [27]. We assumed that the use of the selected optical examination method, despite its shortcomings, would bring improvement in the treatment results for the patients suffering from OSCC in our study. For further investigation, the direct autofluorescence method was chosen.

Direct autofluorescence is a technique used for screening or for better determination of potentially malignant changes of oral mucosa. In 2006, thanks to extensive research efforts, the VELscope system was registered in Canada and also certified by the FDA in the United States. Similar to some other systems working on the principle of natural autofluorescence (Identafi 3000, Sapphire Plus Lesion Detection), the VELscope device uses a non-invasive method of examination using a handpiece emitting bright blue light (400–460 nm), enabling direct visualization of the oral mucosa in real time. This illumination leads to the excitation of endogenous mucosal and submucosal fluorophores [28], which emit a green light, that can be registered through the semipermeable handpiece filter. The visible loss of physiologic fluorescence signifies dysplastic changes of the epithelium, but can be seen also in hyperemia, traumatization, hyperkeratosis, and other benign changes that lower the specificity [29,30,31,32]. In our study, we evaluate the hypothesis that the use of direct autofluorescence (VELscope system tumor mucosal surrounding examination) enhances the ability to reach a pathologically free margin (pFM).

In order to ensure global comparability, and to clearly declare what evaluation criteria were used, it is necessary to properly define the quality of the resection margin. Despite the prevailing belief that leaving part of the tumor cells in a patient’s body is the most common reason for OSCC treatment failure, there is still no clear definition of an adequate resection margin [33,34]. Resection margins are mostly classified as either positive (pPM), that means tumor “cut-through”, close (pCM), or negative (pFM), with different definitions of a healthy tissue rim range [35]. The distance between the tumor border and surgical margin to achieve a pFM varies in some studies [11,36]. According to International Collaboration on Cancer Reporting (ICCR), the definition of a resection margin >5 mm is clear, 1–5 mm is close, and <1 mm is positive [11]. Similar to this definition is a statement from the National Comprehensive Cancer Network (NCCN), where pFM is 5 mm or more from the invasive tumor front, pCM is defined as a distance from the invasive tumor front to the resected margin that is less than 5 mm, and a pPM means a carcinoma in situ or an invasive carcinoma at the margin of resection [37]. Shrinkage of the histological specimen probably plays some role in this uncertainty. There are some studies that address margin shrinkage in patients with head and neck cancer, which was in the order of 20% to 25% [38,39]. In accordance with established international practice, the ICCR model was used to assess the condition of surgical margins.

## 2. Materials and Methods

This pilot prospective randomized study was conducted at the Department of Stomatology, University Hospital Pilsen, Faculty of Medicine in Pilsen, Charles University, Pilsen, Czech Republic, in cooperation with the Sikl’s Department of Pathology, Faculty of Medicine in Pilsen, Charles University, Pilsen, Czech Republic. For better objectivity, the whole study design, data processing, and the results that emerged from the study were discussed with leading European specialists from the University Clinic of Oral and Maxillofacial Surgery, Medical University of Innsbruck, Innsbruck, Austria. We collected two groups (a study group and a random control group) comprising a total number of 122 patients cured in the period 2016–2022 in our department. The inclusion criteria of the study group were as follows: age over 18 years; histologically verified oral squamous cell carcinoma with no sign of inflammation or traumatization of the surrounding mucosa and no previous surgery (except for a small primary biopsy to confirm the diagnosis); radiotherapy or chemotherapy for head and neck cancer; signed informed consent; indication for primary surgical treatment and tumor localization at oral anatomical sites that could be directly visualized using both white light and a fluorescence visualization device (this includes ICD-10 site codes: C02.0-C06.9) [40]; and a VELscope examination before surgery. Other parameters taken into account were tumor site, sex, age, TNM classification, and grade according to the Union for International Cancer Control (UICC). The inclusion criteria of the control group were the same except for the VELscope examination. Simple randomization was used for the distribution of the patients into the above-mentioned groups. All of the patients underwent a standard preoperative examination, including staging based on clinical examination and imaging (ultrasonography, computed tomography, magnetic resonance, or hybrid positron emission tomography). The patients signed a detailed informed consent form in addition to a privacy policy agreement. The design of this study was approved by the Committee of Ethics in Research at the Department of Stomatology, University Hospital Pilsen, Faculty of Medicine in Pilsen, Charles University, under the code 333/2020. This study was conducted according to the Declaration of Helsinki.

Preoperative evaluation of tumor margins with the help of a VELscope device (model No. V1, LED Dental, Inc., 997 Seymour St, Suite 250, Vancouver, BC V6B 3, Canada) was then provided by an experienced surgeon trained and calibrated with the VELscope system for each patient of the study group. It is recommended to provide this examination in a dark room to avoid other illumination interference and to gain the best contrast of the examined field. During this procedure, a field of loss of autofluorescence was marked by permanent (tattoo) or transient (gentian violet—directly before surgery) staining, or marked by a monopolar electrocoagulation device, directly at the beginning of the surgery in general anesthesia (marking modality was chosen according to patient compliance and surgeons’ preference in each case), and the discrepancy between the mark and the daylight-visible tumor boundary was measured (Figure 1). During the examination, it is ideal to direct the excitation beam and thereby observe the fluorescence perpendicular to the mucosal surface. However, even oblique illumination of the mucosa did not cause changes in the range and intensity of fluorescence. Only such tumors were included in the study, which allowed for observation of such angles in direct view, at which the border of fading and natural autofluorescence was still clearly visible.

In an attempt to shorten the surgical time, and thus also the time of general anesthesia for the patient, we developed a technique for permanent tattoo marking, which can be done even a few days before the operation. An insulin syringe was filled with a small amount of conventional tattoo dye, and a little superficial mucosal scratch using the syringe needle applied the dye. For instant specimen orientation and better cooperation with the pathologist, different colors for each specimen site were used. The procedure was performed under topical anesthesia (lidocaine spray 10 g/100 mL) for higher comfort for the patient. Depending on the surgeon’s preference, marking may be performed intraoperatively using electrocoagulation (Figure 2). Interference between the marking and the histological margin examination was avoided with the rim of tissue excised behind the marks, as described subsequently. None of the patients had a peri- or post procedural painful perception, and we noticed no health complications or negative tumor site effects using this procedure.

The intraoperative use of the VELscope device was beneficial only at the beginning of the operation to determine the extent of the resection line, as presented in Figure 2, parts a and b. During the resection, changes in the mucosa around the tumor caused by the surgical trauma no longer allowed a valid evaluation of the loss of fluorescence. Cold steel or a gentle high-frequency electro surgery excision at least 3 mm behind the marks in the study group was performed (in the control group, the excision was performed according to the international conventional recommendation, at least 10 mm behind the naked-eye visible tumor boundary) and the specimen was sent to a dedicated pathologist for histological examination at the Sikl´s Department of Pathology. After hematoxylin-eosin staining and immunohistochemical examination, typing, grading, perineural and intravascular invasion, the tumor to margin distance was evaluated. The histological outcome, VELscope findings, and other data were processed by a professional statistician. The main emphasis was placed on evaluating the increase in the extent of visibility of tumor changes when comparing white light versus the VELscope and, above all, on the differences in the quality of the resection margin of both groups from a histological point of view. As mentioned before, the ICCR model was used for evaluating the status of surgical margins.

## 3. Results

### 3.1. Groups Characteristics

Gender distribution was 40 males and 21 females in the study group, and 36 males and 25 females in the control group, suffering from OSCC (Table 1). The Fisher´s exact test with a *p*-value of 0.5754 showed no significant difference between the two groups.

The mean age in the study group was 65.3 years, ranging from 37 to 90 years of age, and the mean age was 64.3 years, ranging from 34 to 88 years, in the control group ([Fig diagnostics-13-03161-ch002]). The similar age distribution of both groups is demonstrated by a *p*-value of 0.7845, according to the Wilcoxon Two Sample test.

The most affected site of the oral cavity was the tongue in both groups (Table 2). According to a *p*-value of 0.6183 using the Chi-Square test, we did not notice any statistically significant difference.

The stage and grade status of all tumors were evaluated, and the situation in both groups is shown in their respective tables (Table 3 and Table 4). In both groups, initial stages were most frequently represented, but stage IVa was also present in a large percentage. According to the results of histological examination, this phenomenon is caused by early invasion of the jaw bone near the tumor. To evaluate the similarity of the two groups, the Chi-square test was used according to stage and grade, with a *p*-value of 0.1472 and 0.3587, respectively.

Statistical analysis did not reveal any significant difference between the two groups in terms of the features described above.

### 3.2. Treatment Outcomes Comparison

The loss of physiologic fluorescence resulted in a resection enlargement of 4.68 mm on average (1–12 mm) compared to the polychromatic light and the tactile tumor borders assessment.

The histological examination revealed no case of pPM (*n* = 0; 0%), six cases of pCM (*n* = 6; 9.84%), and fifty-five specimens were pFM (*n* = 55; 90.16%) in the mucosal margins of the study group. The situation in the control group was as follows: pPM (*n* = 7; 11.48%), pCM (*n* = 14; 22.95%), and pFM (*n* = 40; 65.57%) ([Fig diagnostics-13-03161-ch003]). Although the autofluorescence technique had no influence on the deep surgical margins, we present the deep margin situation for comprehensiveness. In the study group we encountered pPM in eight cases (*n* = 8; 13.11%), pCM in ten cases (*n* = 10; 16.39%), and pFM in forty-three cases (*n* = 43; 70.49%); in the control group the results were as follows: pPM (9; 14.75%), pCM (6; 9.84%), and pFM (46; 75.41%) ([Fig diagnostics-13-03161-ch004]). We did not observe any statistically significant differences in the quality of the resection margin in relation to the different anatomic locations of the primary tumor origin.

### 3.3. Observed Effect of Autofluorescence Assistance

The results of our observation are summarized in the attached table (Table 5). The H0 hypothesis was established that there is no difference in the ratio of MpFM (mucosal pathological free margin) to MpCM+MpPM (mucosal pathological close and positive margin) between the two groups. This hypothesis was tested using Fisher’s exact test, and with a resulting *p*-value of 0.019, the H0 hypothesis was rejected. Thus, a statistically significant difference between the two groups was demonstrated. The risk of the presence of close or positive lateral margins was thus 4.8 times higher in the control group than in the study group. There was no significant difference in deep margins. Despite the larger resection in the study group, there was no significant increase in postoperative morbidity with regard to either swallowing or speech. No patient experienced major complications in the postoperative period, and no patient was discharged with a feeding tube or tracheostomy. The overall treatment outcome, in terms of surgical tumor eradication, was significantly better in the study group compared to the control group.

## 4. Discussion

A multimodal treatment with an accent on radical surgical tumor excision has the best curative results [41,42]. Early detection, prompt staging, and individual setting of the required treatment plan can lower the risk of curative failure, recurrence, and enormous impairment of the quality of life of the patient [4]. New examination techniques, as well as modern surgical approaches, came into clinical practice thanks to scientific progress [43]. One of the modern techniques described in our study—mucosal autofluorescence—has the potential to help clinicians in early detection of mucosal malignity in the oral cavity, and, in addition, to determine the tumor boundaries more precisely [12,44,45,46]. The surgeon’s main goal in the surgical treatment of OSCC is to achieve an R0 resection, which means ideally the pFM [42]. The presence of residual tumor cells is believed to be the most important prognostic factor. A large number of studies comparing local recurrence rates with margin status in OSCC found a strong correlation, although the absolute number of local recurrences and the criteria used to define positive margins vary significantly among particular studies [47,48,49]. Some studies surprisingly failed to demonstrate any correlation between recurrence and margin status [50]. It is still not clear how wide the distance between surgical margin and tumor should be. It is crucial to obtain adequate surgical margins around the tumor, however, the surgeon must find a balance between the radicality of the operation and efforts to preserve the function of the orofacial system; they must preserve quality of life and limit cosmetic disability [3,51,52,53,54,55,56,57]. Many scientific publications deal with the problem of positive resection margins, unfortunately there are currently no globally valid guidelines. The setting of a minimum safety margin is discussed in a number of studies, and the range varies between 10–2 mm; the median recommended clinical distance for the resection of OSCC is 5 mm, although a clear basis for this distance is currently lacking. Mainly because of specimen shrinkage and “invisible” carcinoma cell spread, the excised rim of macroscopically unchanged mucosa should be more than the above-mentioned pFM distance. It is recommended to keep at least a 10 mm distance from the tumor to avoid impairing the histological margin status during the surgery [58]. The intraoperative evaluation of resection margins can determined using several methods. The use of frozen sections (FS) used to be the gold standard; it was accepted worldwide but, as underlined by Yahalom et al. [59], the procedure is not standardized. Another topic of discussion is the site where the FS should be taken, if it is the resection bed or the specimen margin [60]. Another problem is the risk of incorrectly identifying the area to be re-excised in the event of positive margins [61] and time-consumption during surgery. For this reason, the endeavor of finding a more favorable method for intraoperative determination of adequate resection margins is very topical, and research is very dynamic in this regard. Some more methods have been suggested for better achieving pFM in mucosal layer OSCC surgical treatment or oral mucosa malignity detection, and we present a list of some of them for comparison below:

Contact endoscopy—allows an in vivo microscopic examination of upper aerodigestive tract mucosa with a rigid endoscope. It is a noninvasive technique and provides information on microscopic diagnosis and lesion margins. A sensitivity of 80%, a specificity of 100%, and an accuracy of 93% for contact endoscopy in the diagnosis of malignancy is reported [62,63,64].

Narrow band imaging is a video endoscopic system for the examination of mucous membranes. Thanks to narrow band filters, only two specific bands of visible light, which are typical for the absorption peak of hemoglobin, are allowed to pass through. The observed wavelengths increase the visibility of microvascular abnormalities that could be related to preneoplastic and neoplastic mucosal changes [65,66,67,68,69].

Staining with Lugol’s iodine solution or toluidine blue is one of the methods that can reduce the number of positive margins by pointing out the tumor margins [70,71,72,73,74]. Vital staining utilizes the enhanced affinity of some dyes to certain cell structures present in dysplastic or malignant cells, making them more apparent. The widest use has toluidine blue in the oral cavity mucosa. This basic metachromatic stain has an affinity toward DNA and RNA and illustrates an invasive malignancy, carcinoma in situ (CIS), or dysplasia by staining abnormal tissues blue [74,75].

Using touch imprint cytology, the properties of the cells located on the surface of the resection margin can be evaluated intraoperatively. It is a fast, simple, cheap, and relatively accurate examination technique [76].

Optical coherence tomography is another technique for distinguishing between positive and negative surgical margins [77,78].

Confocal laser endomicroscopy (CLE) is a method which has proven its worth in recent years, especially in intraoperative incision margin evaluation, where it allows in vivo cell imaging [79,80].

Some publications mention a molecular definition of surgical margins, from protein markers to DNA-based techniques, that can evaluate the margins at the subcellular level and can explain, for example, local recurrence in optical microscope pFM [81,82,83]. Various monoclonal antibodies and other ligands are already being clinically used or researched in this area, which have the ability to mark the structures of tumor cells in a certain way and make them visible intraoperatively, for example, by means of fluorescence [14,15,19,20].

The final technique is tissue reflectance. The ViziLite device, which uses reflectance, is adapted for use in the oral cavity. The principle of this method is that the abnormal cells reflect the light (high nucleus–cytoplasm ratio, keratinization excess, hyperparakeratinization), while normal cells absorb the light and are depicted in a bluish color [32].

We verified the potential of autofluorescence to detect dysplastic and malignant mucosal changes in our study. The limitations of this technique and of our methodology were that the detection capability stayed only on the superficial mucosal layer. In our study, we identified some possible sources of bias that need to be taken into account. A frequent weakness of clinical studies is that the compared groups are not identical but statistically very similar, which is also the case in our study. Another problem is the burden of the natural autofluorescence method, which has a relatively low specificity despite a very high sensitivity. We partially eliminated this weakness by the fact that the histological nature of the disease was known and further by using the set inclusion parameters. According to published studies, the above-mentioned alternative optical examination techniques also achieved promising results, but there is a lack of scientific work relevantly comparing individual methods with each other; further research is needed in this area. The biggest limitation of autofluorescence as a technique is its limitation to the mucosal layer. In this regard, it would be appropriate to combine the method with another similarly non-burdensome and simple examination technique that detects deep tumor margins. Such a technique could be, for example, the adjuvant perioperative use of ultrasonography, which also achieves very good results [21,22,23]. In this sphere, we consider further research to be extremely interesting, as the potential of both methods could be exponentially enhanced. Another limitation of the technique is due to the concept of the device, where the handpiece is held by the examiner’s hand, allowing examination only with a direct view. This means that tumors with an unfavorable anatomical location for examination with a direct view could not be treated with this method. In some cases, the fading of fluorescence in the location of the dorsum of the tongue was masked by the presence of bacterially colonized hypertrophically keratinized filiform papillae, which fluoresce bright orange-red. This problem was easily solved by gentle mechanical removal of this layer before the examination, for example, using a tongue scraper. On the other hand, we developed a technique of preoperative marking of the pathological mucosal changes using a permanent tattoo, which can help the surgeon to set the resection line more precisely. We gained a better specimen orientation, which led to surgical treatment success enhancement and more precise orientation, for better pathologist communication and better identification of risk field in case of re-excision needed by this technique. Another advantage of direct autofluorescence is a low cost burden, which is approximately USD 2 per examination; the purchase price of the device, as a one-time investment, varies between USD 1695 and USD 1995 according to the manufacturer’s recommendations.

The potential of this technique for better detection of mucosal malignancy is well known and there have been multiple published articles. The role of autofluorescence in setting the margins into the R0 region, or even better, in achieving pFM, is being researched worldwide. The results of some studies are promising, but it is hard to compare the results because of its nonhomogenity. Some presented works are limited to the early stage of OSCC only, or do not depict the resection margin changes clearly [12,46,84,85]. Although this technique is limited to the mucosal layer, some studies report a reduction in local recurrence of more than 30% [12]. Further investigation in this field is needed to assess the potential of this technique and to create a recommendation for the standardized use of autofluorescence in OSCC surgery as a guideline.

## 5. Conclusions

Our findings support the hypothesis of improving the surgery outcomes in the study group, and we discovered a statistically significant difference between the study and the control group. We proved the efficacy of mucosal autofluorescence in setting sufficient mucosal surgical margins in our study. If combined with the permanent tattoo marking of the pathological changes visible under VELscope, as described above, this technique can enhance the surgeon´s ability to successfully treat a patient with OSCC even more.

## Data Availability

The data presented in this study are available on request from the corresponding author. The data are not publicly available due to institutional committee of ethics statement.

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
