# Peer review of "Clinical Experience with Autofluorescence Guided Oral Squamous Cell Carcinoma Surgery"

_diagnostics, 2023, doi:10.3390/diagnostics13203161_

Round 1
Reviewer 1 Report (Previous Reviewer 1)
The authors have adequately addressed all points. I agree with the publication in its present form.
Author Response
Dear reviewer, thank you very much for your encouraging comment. We are very pleased that after editing the manuscript after the first review round, the article is at such a level that, in your opinion, no further adjustments or additions are needed and it is possible to use the current version of the text for publication. We appreciate your feedback and thank you very much for taking the time to review our study.
Reviewer 2 Report (New Reviewer)
Thank you for the opportunity to review this manuscript. The issue addressed is interesting and of high clinical relevance, the manuscript is clearly structured and well written.
After also considering the first round of reviews and the profound answers the authors provided to the other reviewers, only a small amount of my questions was left unanswered.
1. While the large number of tumors located at the tongue may possibly impede this analysis, I still have to ask: Did you see any differences in resection margins depending on tumor localization? I feel that this would be possible due to the different accessibility to the surgeon as well as different shrinkage of tissue in dependence of origin (e.g. tongue vs. hard palate).
2. You write this was a retrospective study. As patients were randomized, I would suspect a prospective setting – could you clarify this for me?
3. You included patients from 2016 to 2022 – a period long enough to suspect an effect on survival. Did you conduct a survival analysis (local control/local recurrences for instance) to evaluate a possible oncological effect of your autofluorescence marking – and if not, why not?
4. You clearly present better margins in the study group (4.8 times etc.). Do you consider all margins (including basal margins that are not affected by autofluorescence as you state yourself) in this statement or only lateral margins?
5. As far as I am familiar with intraoral autofluoresence guided surgery, apart from blood stained surfaces etc., also angulation plays a relevant role. While the tongue may allow for manipulation to provide a flat surface and optimal angulation, I would imagine that tumors in the floor of the mouth reaching into the distal areas or also affecting the lingual mandibular area may be hard to evaluate. Could you formulate a statement regarding this possible limitation?
Author Response
Dear reviewer, Your comments and questions are factual and have helped us identify several inaccuracies and possible sources of misinterpretation. Also, thanks to you, we have added some explanatory information about performing natural autofluorescence investigations. Thank you very much for your comments and the time devoted to evaluating our work. We appreciate it very much.
- Comment: While the large number of tumors located at the tongue may possibly impede this analysis, I still have to ask: Did you see any differences in resection margins depending on tumor localization? I feel that this would be possible due to the different accessibility to the surgeon as well as different shrinkage of tissue in dependence of origin (e.g. tongue vs. hard palate).
Response: In our study, we did not record a statistical association between the primary location of the tumor and the quality of the resection margin, nor were there any differences in the extent of fluorescence fading around the tumor between individual locations. Tumors that could not be adequately examined by direct observation due to unfavorable anatomic conditions were not included in the study. During the marking of the border of fluorescent fading around the tumor during the examination with the VELscope device, in areas where there are mucosal folds or the mucosa is collapsed, the tissue is stretched and directed with assistance in such a way that it is possible to safely perform the examination using autofluorescence in direct view. In the area of the hard and soft palate, this is not necessary and it is enough just to move the tongue out of sight. The force required to stretch the mucosa in order to be able to carry out the aforementioned examination may vary with regard to the rigidity of the tissues around the infiltrating tumor and its location, and it is not possible to objectify it sufficiently precisely. Clarity of the examined field is governed by the needs of the surgeon performing the marking of the lesion and the subjective feedback of the visual and tactile sensations of the assistance. In the course of our research, we did not notice that in some localities a greater extent of lack of fluorescence was found around the tumor than in other localities of the primary origin of the tumor. Only in the area of the dorsum of the tongue with hypertrophy of the filiform papillae and their excessive colonization by bacteria they do fluoresce orange and thus mask the fading of the fluorescence. This problem is easily solved by wiping off the contaminated keratinized layer of the papillae. Only patients in whom it was possible to safely examine the area around the tumor with a direct view were included in our research. We were unable to find in the literature or in the manufacturer's recommendations the angular limit of the beam falling on the mucous membrane, up to which such an examination can be safely performed. From our empirical experience, no difference in the extent of fluorescence fading was observed when the examined area was illuminated at different angles. Even at a relatively oblique viewing angle, the borders of the fade were clearly visible. However, tumors affecting the radix of the tongue could no longer be ideally examined with a direct view, and we excluded such patients from the study. We made adjustments to the Results (8th page, line 229 - 231) and Discussion section (11th page, line 345 – 352) in the article based on your comment.
- Comment: You write this was a retrospective study. As patients were randomized, I would suspect a prospective setting – could you clarify this for me?
Response: Your assumption is correct, patients were randomized before starting treatment and the study concept was established before data collection and evaluation. The data were retrospectively examined and statistically processed, however, the conduct of our research meets the characteristics of a prospective study. This inaccuracy was corrected in the article, thank you for your observation and logical check, and I apologize for the primarily inaccurate naming of the type of study. Correction made in the article at the section Materials and Methods, 3rd page, line 97.
- Comment: You included patients from 2016 to 2022 – a period long enough to suspect an effect on survival. Did you conduct a survival analysis (local control/local recurrences for instance) to evaluate a possible oncological effect of your autofluorescence marking – and if not, why not?
Response: Our thoughts are going exactly in this direction and we would like to obtain a larger number of patients, included in our study, who will reach a time interval of five years from surgical therapy. After that, we will be able to statistically process a sufficiently large set of patients including information on, for example, disease-free survival, overall survival and specific overall survival. We believe that we will be able to collect the required data within two years and we will present the result in a subsequent publication. Not mentioned in the article, we hope that our explanation and promise of subsequent publication will be sufficient for you.
- Comment: You clearly present better margins in the study group (4.8 times etc.). Do you consider all margins (including basal margins that are not affected by autofluorescence as you state yourself) in this statement or only lateral margins?
Response: Thank you for this comment, our work was focused on the mucosal layer and the mentioned difference describes the situation only in the plane of the lateral margins. There was no statistically significant difference between the two groups in the area of the deep margins. We made an adjustment in the manuscript (Abstract, 1st page, line 30 and Results, 9th page, line 243) so that the results of our work could not be interpreted in a confusing way.
- Comment: As far as I am familiar with intraoral autofluoresence guided surgery, apart from blood stained surfaces etc., also angulation plays a relevant role. While the tongue may allow for manipulation to provide a flat surface and optimal angulation, I would imagine that tumors in the floor of the mouth reaching into the distal areas or also affecting the lingual mandibular area may be hard to evaluate. Could you formulate a statement regarding this possible limitation?
Response: This problem is already partially described in the reply to comment number one. There are clearly locations in the oral cavity that are both easier and more difficult to examine with a direct view. Examination by means of natural autofluorescence is fundamentally no different from examination with the naked eye. That means that the locations accessible to examination by a direct view are also accessible to examination using autofluorescence. It is not always possible to illuminate the examined area and examine it at an angle of 90 degrees, however, a slight optical angulation does not play a role in the examination result. As mentioned earlier, the limit angle for observation of the mucosal surface by means of the VELscope has not yet been defined. In the environment of the cells of the mucous membrane, scattering, absorption and reflection of the beam occur during any illumination. The closer we get to 90 degrees in our investigation, the better the excitation versus reflection ratio will be. The Velscope's semitransparent filter co-filters the reflected light and passes the wavelengths emitted by the excised fluorophores. In a simplified way, it would be possible to formulate a recommendation for the investigation that valid results can only be achieved at such observation angles, in which the border of lack of fluorescence and the natural greenish fluorescence of the healthy mucosa can be safely distinguished. In cases where it is not possible to examine the tumor surroundings completely and with sufficient quality with a direct view, the technique of autofluorescence assisted surgery should not be used. The statement was included into our manuscript. Changes made in Materials and Methods, 3rd page, lines 134 – 139 and Discussion, 11th page, lines 345 – 352.
This manuscript is a resubmission of an earlier submission. The following is a list of the peer review reports and author responses from that submission.
Round 1
Reviewer 1 Report
Dear authors,
Thank you very much for the opportunity to review this excellent paper. It is a very clean elaboration of a procedure that shows promise in clinical application. The paper is methodologically good and very clearly written.
However, a few essential points should be addressed:
1) In the introduction, the authors write, "New effective examination and imaging techniques are being developed that allow 46 the surgeon to better visualize the boundaries of the primary tumor before and during the 47 surgery more precisely [12-24]." Unfortunately, the sources do not mention confocal laser endomicroscopy (CLE). However, this method has proven its worth in recent years, especially in intraoperative incision margin control applications. Two essential publications that should be mentioned in this context are:
Sievert M, Stelzle F, Aubreville M, Mueller SK, Eckstein M, Oetter N, Maier A, Mantsopoulos K, Iro H, Goncalves M. Intraoperative free margins assessment of oropharyngeal squamous cell carcinoma with confocal laser endomicroscopy: a pilot study. Eur Arch Otorhinolaryngol. 2021 Nov;278(11):4433-4439. doi: 10.1007/s00405-021-06659-y. Epub 2021 Feb 13. PMID: 33582849; PMCID: PMC8486707.
AND
Sievert M, Oetter N, Mantsopoulos K, Gostian AO, Mueller SK, Koch M, Balk M, Thimsen V, Stelzle F, Eckstein M, Iro H, Goncalves M. Systematic classification of confocal laser endomicroscopy for the diagnosis of oral cavity carcinoma. Oral Oncol. 2022 Sep;132:105978. doi: 10.1016/j.oraloncology.2022.105978. epub 2022 Jun 21. PMID: 35749803.
CLE is also a promising method for in vivo imaging of cells. If NBI is listed in the discussion, CLE should also be discussed in a paragraph.
2) Figure 1: Image quality should be improved—higher resolution.
3) Table 1-3: It would be desirable to provide a p-value regarding the distribution of individual parameters between the two groups (study and control) (Chi-square test) to demonstrate that there are no statistically significant differences between the two cohorts.
4) Please mention the cost of this method in the discussion section.
After correcting the points I mentioned, I consider the manuscript's publication.
With kind regards.
Author Response
Dear Reviewer, thank you very much for evaluating our forthcoming publication. Your observations and comments are factual, professional and clearly lead to an increase in the quality of the work. With pleasure I accepted all your recommendations and adjusted the article accordingly. I would like to add comments on individual points below.
Point 1) In the introduction, the authors write, "New effective examination and imaging techniques are being developed that allow 46 the surgeon to better visualize the boundaries of the primary tumour before and during the 47 surgery more precisely [12-24]." Unfortunately, the sources do not mention confocal laser endomicroscopy (CLE). However, this method has proven its worth in recent years, especially in intraoperative incision margin control applications. Two essential publications that should be mentioned in this context are:
Sievert M, Stelzle F, Aubreville M, Mueller SK, Eckstein M, Oetter N, Maier A, Mantsopoulos K, Iro H, Goncalves M. Intraoperative free margins assessment of oropharyngeal squamous cell carcinoma with confocal laser endomicroscopy: a pilot study. Eur Arch Otorhinolaryngol. 2021 Nov;278(11):4433-4439. doi: 10.1007/s00405-021-06659-y. Epub 2021 Feb 13. PMID: 33582849; PMCID: PMC8486707.
AND
Sievert M, Oetter N, Mantsopoulos K, Gostian AO, Mueller SK, Koch M, Balk M, Thimsen V, Stelzle F, Eckstein M, Iro H, Goncalves M. Systematic classification of confocal laser endomicroscopy for the diagnosis of oral cavity carcinoma. Oral Oncol. 2022 Sep;132:105978. doi: 10.1016/j.oraloncology.2022.105978. epub 2022 Jun 21. PMID: 35749803.
CLE is also a promising method for in vivo imaging of cells. If NBI is listed in the discussion, CLE should also be discussed in a paragraph.
Response 1)
The technique of confocal microscopy is well known, the first practically usable confocal microscope was developed in our country, and it is a pleasure to watch the progress of this method. On the other hand, it is a shame I hadn´t mentioned it in our article. Thank you very much for your reminder. I deeply regret that I missed the scientifically interesting articles recommended by you that are very close to my topic in the preparation of the publication. Please excuse this shortcoming, both of your articles were properly cited in the text, the CLE method was briefly mentioned in the discussion. I would like to keep the text clear and not discuss the other mentioned methods in detail, but if you want the specific details of the method to be described, it is not a problem to give more space to this interesting method within reason.
Point 2) Figure 1: Image quality should be improved—higher resolution.
Response 2) Resolution of the Figure 1 was enhanced.
Point 3) Table 1-3: It would be desirable to provide a p-value regarding the distribution of individual parameters between the two groups (study and control) (Chi-square test) to demonstrate that there are no statistically significant differences between the two cohorts.
Response 3) The statistical evaluation of the whole study is quite extensive, it has been provided by our statistician and to be honest, I am not an expert in this field. I added a Chi-Square test and Wilcoxon Two Sample test results for the tables 1 – 4, it means gender, age, tumour side, grade and stage. I hope this would be sufficient and clear. In case of uncertainties I will improve it with the help of our expert.
Point 4) Please mention the cost of this method in the discussion section.
Response 4) The costs of this method were added to discussion. Costs may vary slightly in individual countries and workplaces, depending on the insurance system, currency, composition of the medical team, etc., so I have included the manufacturer's information.
Reviewer 2 Report
The investigators have attempted to study a non - invasive , imaging as tool to delineate tumour margins in oral squamous cell carcinoma . However , the study design need clarification on the following
1. Evaluating depth of invasion is very vital while resecting the tumour. With the understanding that the imaging only the surface by autofluorescence needs to be justified. Will this be an adjunct and not confirmatory ?
2. Stage of tumour and sub- site are important is assessing , which is not considered here
3.Keratinisation of the mucosa has a significant role is determining the loss of fluorescence , which is not considered here
4. To consider autofluorescence imaging as a intra-operative tumour margin tool is not conclusive
Thanks
Author Response
Dear reviewer,
Thank you very much for your comments, I would like to consider all the points below. I'm not sure if I understood all your admonitions correctly. In case you are not satisfied with my answers, please state the question of uncertainty clearly. It would be our pleasure to fulfill your recommendation.
Best wishes
Point 1) Evaluating depth of invasion is very vital while resecting the tumour. With the understanding that the imaging only the surface by autofluorescence needs to be justified. Will this be an adjunct and not confirmatory ?
Response 1) The subject of our research was to prove the hypothesis that although the method of direct autofluorescence is limited to the mucous layer, its use will lead to an increase in the success of the treatment. We are fully aware of this shortcoming and I hope the article shows it. “We verified the potential of autofluorescence to detect the dysplastic and malignant mucosal changes in our study. The limitations of this technique and the methodology of our study is that the detection capability stays only on the superficial mucosal layer.” “The biggest limitation of autofluorescence in the way of use we are investigating is the limitation to the mucosal layer. In this regard, it would be appropriate to combine the method with another similarly non-burdensome and simple examination technique that detects deep tumor margins. Such a technique could be, for example, the adjuvant perioperative use of ultrasonography, which also achieves very good results [16–18]. In this sphere, we consider further research to be extremely interesting, as the potential of both methods could be exponentially enhanced.”
Point 2) Stage of tumour and sub- site are important is assessing , which is not considered here
Response 2) We present a stage of the tumor in a table 4:
Table 4. Tumor stage distribution in both groups
|
Stage |
Frequency – amount / percentile |
Total |
|
|
Study group |
Control group |
||
|
I |
16 / 26.23 |
8 / 13.11 |
24 |
|
II |
15 / 24.59 |
14 / 22.95 |
29 |
|
III |
6 / 9.84 |
13 / 21.31 |
19 |
|
Iva |
22 / 36.07 |
23 / 37.70 |
45 |
|
IVb |
1 / 1.64 |
3 / 4.92 |
4 |
|
IVc |
1 / 1.64 |
0 / 0.00 |
1 |
|
Total |
61 |
61 |
122 |
According to the recommendation to the other reviewer, we also present a statistical evaluation of the two groups: The stage and grade status of all tumors were evaluated and the situation in both groups is presented in appropriate tables (Table 3, 4). The Chi-Square test was used to evaluate the similarity of both groups.
Grade - Chi-Square Stage - Chi-Square
DF Value Prob. DF Value Prob.
2 2.0503 0.3587 5 7.3023 0.1991
A tumor sub site is presented in table 2:
Table 2. The distribution of tumor origin in the oral cavity mucosa.
|
Diagnosis |
Frequency – amount / percentile |
Total |
|
|
Study group |
Control group |
||
|
C02 |
19 / 31.15 |
21 / 34.43 |
40 |
|
C03 |
18 / 29.51 |
14 / 22.95 |
32 |
|
C04 |
18 / 29.51 |
18 / 29.51 |
36 |
|
C05 |
2 / 3.28 |
1 / 1.64 |
3 |
|
C06 |
3 / 4.92 |
7 / 11.48 |
10 |
|
C09 |
1 / 1.64 |
0 / 0.00 |
1 |
|
Total |
61 |
61 |
122 |
In the section Materials and Methods is posted, which tumor sites were included to the study:
The inclusion criteria of the study group were as follows: age over 18 years, histologically verified oral squamous cell carcinoma with no sign of inflammation or traumatization of the surrounding mucosa and no previous surgery (except for a small primary biopsy to confirm the diagnosis), radiotherapy or chemotherapy for head and neck cancer, signed informed consent, indication for primary surgical treatment and tumor localization at oral anatomical sites that can be directly visualized using both white light and fluorescence visualization device (this includes ICD‐10 site codes: C02.0‐C06.9)[38] and VELScope examination before surgery.
Point 3).Keratinisation of the mucosa has a significant role is determining the loss of fluorescence , which is not considered here
Response 3) There are more factors that affects the specificity of this method, some of them (Keratinization among them) are mentioned in Introduction:
This illumination leads to the excitation of endogenous mucosal and submucosal fluorophores [26], which emit a green light, that could be registered through the semipermeable handpiece filter. The visible loss of physiologic fluorescence signifies dysplastic changes of the epithelium, but could be seen also in hyperemia, traumatization, hyperkeratosis and other benign changes, that lower the specificity [27–30].
Point 4). To consider autofluorescence imaging as a intra-operative tumour margin tool is not conclusive
Response 4) The intraoperative use of autofluorescence has been published and the outcomes are promising as seen in the presented citations:
The potential of this technique for better detection of mucosal malignancy is well known and have been multiple published. The role of autofluorescence in setting the margins into the R0 region or even better in achieving the pFM is being worldwide researched. The results of some studies are promising, but it is hard to compare the results because of its nonhomogenity. Some presented works are limited to the early stage of OSCC only, or do not depict the resection margin changes clearly [44,45,83–85]